# Predicting mortality from AI cardiac volumes mass and coronary calcium on chest computed tomography

Robert J. H. Miller[1,2,8], Aditya Killekar [1,8], Aakash Shanbhag[1], Bryan Bednarski[1], Anna M. Michalowska[1], Terrence D. Ruddy [3], Andrew J. Einstein[4,5], David E. Newby [6], Mark Lemley [1], Konrad Pieszko[1,7], Serge D. Van Kriekinge [1], Paul B. Kavanagh[1], Joanna X. Liang[1], Cathleen Huang[1], Damini Dey [1], Daniel S. Berman [1] & Piotr J. Slomka [1] ✉

Chest computed tomography is one of the most common diagnostic tests, with 15 million scans performed annually in the United States. Coronary calcium can be visualized on these scans, but other measures of cardiac risk such as atrial and ventricular volumes have classically required administration of contrast. Here we show that a fully automated pipeline, incorporating two artificial intelligence models, automatically quantifies coronary calcium, left atrial volume, left ventricular mass, and other cardiac chamber volumes in 29,687 patients from three cohorts. The model processes chamber volumes and coronary artery calcium with an end-to-end time of ~18 s, while failing to segment only 0.1% of cases. Coronary calcium, left atrial volume, and left ventricular mass index are independently associated with all-cause and cardiovascular mortality and significantly improve risk classification compared to identification of abnormalities by a radiologist. This automated approach can be integrated into clinical workflows to improve identification of abnormalities and risk stratification, allowing physicians to improve clinical decision-making.

Coronary artery disease (CAD) is a leading cause of morbidity and mortality[1,2]. Coronary artery calcium (CAC) scores obtained from non-contrast ECG-gated computed tomography (CT) has emerged as a method for evaluation of asymptomatic patients[1,2]. CAC scores are a robust marker of cardiovascular risk[3–10], and may even help improve patient compliance with medical therapies and lifestyle interventions[11,12]. Contrast-enhanced cardiac CT also provides information regarding cardiac chamber volumes and left ventricular (LV) mass which are predictive of mortality[13], and cardiovascular events[14]. Importantly, CAC is not routinely evaluated on non-cardiac CT.

Additionally, cardiac chamber volumes and left ventricular mass classically could not be evaluated on non-contrast CT, since contrast is required to differentiate myocardium from blood pool and to identify the valve planes which separate cardiac chambers. However, these non-cardiac, non-contrast CT scans make up the vast majority of the over 15 million chest CT scans performed annually in the United States alone[15].

Recent advancements in artificial intelligence (AI) have potentially enabled quantification of these measures from non-gated CT imaging. CAC can be manually measured from non-gated CT imaging[16], with

[1]Departments of Medicine (Division of Artificial Intelligence in Medicine), Imaging and Biomedical Sciences Cedars-Sinai Medical Center, Los Angeles, CA, USA. [2]Department of Cardiac Sciences, University of Calgary, Calgary, AB, Canada. [3]Division of Cardiology, University of Ottawa Heart Institute, Ottawa, Ontario, Canada. [4]Division of Cardiology, Department of Medicine, Columbia University Irving Medical Center and New York-Presbyterian Hospital, New York, New York, NY, USA. [5]Department of Radiology, Columbia University Irving Medical Center and New York-Presbyterian Hospital, New York, New York, NY, USA. [6]British Heart Foundation Centre for Cardiovascular Science, University of Edinburgh, Edinburgh, UK. [7]Department of Interventional Cardiology and Cardiac Surgery, University of Zielona Gora, Gora, Poland. [8]These authors contributed equally: Robert J. H. Miller, Aditya Killekar. ✉e-mail: Piotr.Slomka@cshs.org

excellent correlations CAC scores from gated examinations[17]. However, manual annotation of CAC is time consuming, particularly for the lower radiation dose scans, which are used for cancer screening. AI has been applied to automate quantification of CAC from lung cancer screening CT scans[18,19], and was associated with cardiovascular mortality in a selected cohort of patients from the National Lung Screening Trial (NLST)[20]. A recently developed AI model may also facilitate quantification of left and right atrial and ventricular volumes and LV mass from non-contrast CT[21], but these estimates from ungated, non-contrast CT have never been validated as markers of risk.

We integrated our convolutional AI model which automatically measures CAC[22–24], with another AI model (TotalSegmentator) which automatically segments cardiac chamber volumes[21]. The aim of our study was to evaluate the clinical potential of a fully automated AI pipeline that estimates CAC, cardiac chamber volumes, LV mass, and shape index when applied to low-dose (non-contrast and ungated) lung CT with respect to predicting clinical outcomes in three external populations.

## Results

### Population characteristics—NLST

We included a total of 24354 patients with median age 61 (IQR 57–65), of whom and 14441 (59.3%) were males. The overall study design is shown in Fig. 1. The model was able to process chamber volumes and coronary artery calcium with an end-to-end processing time of ~18 s, while failing to segment only 0.1% of cases. In total, 4618 (19.0%) had CAC 0, 9006 (37.0%) had CAC 1–100, 4816 (19.8%) had CAC 101–400, and 5914 (24.3%) had CAC > 400. Population characteristics in patients categorized by extent of CAC are presented in Table 1.

Histograms outlining the distribution of CAC, LV, left atrial (LA), right ventricular (RV), and right atrial (RA) volumes and LV mass index are shown in Supplemental Fig. 1 and the correlation between values as shown in Supplemental Fig. 2. Correlation between gated CT measurements and ungated CT estimates are summarized in Supplemental Fig. 3. All Spearman correlations were excellent (LV myocardium

$r = 0.947$, LA volume $r = 0.926$, RA volume $r = 0.893$, LV volume $r = 0.793$, and RV volume $r = 0.922$). Comparisons between baseline CT estimates and one-year CT estimates in 22292 patients are shown in Fig. 2. Correlations were excellent (LV myocardium $r = 0.917$, LA volume $r = 0.866$, RA volume $r = 0.864$, LV volume $r = 0.892$, and RV volume $r = 0.899$) with no significant bias.

Case examples showing segmentation of CAC and chamber volumes are shown in Fig. 3. The current clinical standard, radiologist identified cardiovascular abnormalities, were noted in a minority of scans, with reported abnormalities on 62 (1.3%) patients with CAC 0, 311 (3.5%) patients with CAC 1–100, 360 (7.5%) patients with CAC 101–400, and 828 (14.0%) patients with CAC > 400 and in only 303 (9.3%) of patients with abnormal LV mass index.

### Associations with all-cause mortality—NLST

During median follow-up of 6.7 years (IQR 6.3–7.0), 1795 (7.4%) patients died. Of those deaths, 459 (25.6%) were adjudicated as cardiovascular deaths. Kaplan-Meier curves for all-cause mortality stratified by CAC categories are displayed in Supplemental Fig. 4. Increasing CAC category was associated with an increasing risk of all-cause mortality as shown in Supplemental Table 1. Identification of cardiovascular abnormality by the radiologist was associated with less risk (unadjusted HR 1.84, 95% CI 1.58–2.13) than the presence of CAC > 100. Quartiles of LV, LA, RV, and RV volume provided risk stratification for all-cause mortality as shown in Supplemental Fig. 5. Abnormal LV mass index was also associated with an increased risk of mortality (unadjusted HR 1.76, 95% CI 1.57–1.97).

Associations with all-cause mortality in the multivariable model are presented in Supplemental Table 2. Patients with CAC 1–100 (adjusted HR 1.24, 95% CI 1.04 –1.47), CAC 101–400 (adjusted HR 1.56, 95% CI 1.30–1.87), and CAC > 400 (adjusted HR 1.88, 95% CI 1.57–2.24) were at increased risk of all-cause mortality. Increasing LA volume (adjusted HR 1.11, 95% CI 1.06–1.16), LV mass index (adjusted HR 1.34, 95% CI 1.22–1.47), and shape index (adjusted HR 1.31, 95% CI 1.02–1.66) were also associated with increased risk of death.

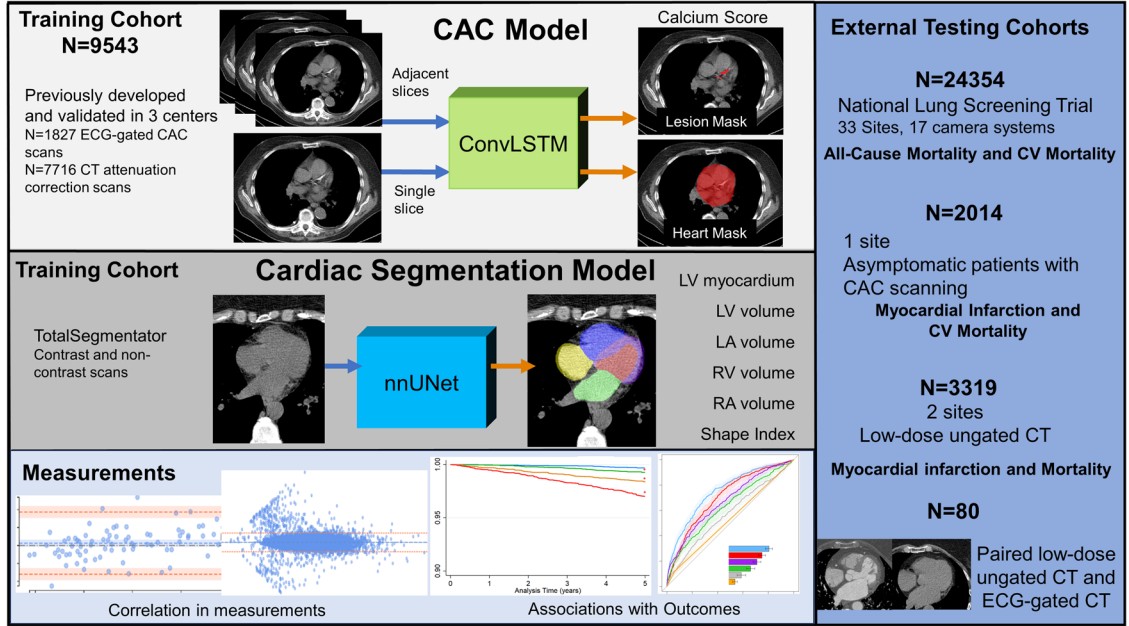

**Fig. 1 | Overall study design.** We utilized a convolutional long short-term memory (ConvLSTM) model (light gray) which uses computed tomography slices as inputs (blue arrows) to segment coronary artery calcium (CAC). TotalSegmentator (dark gray), which uses a no new-net UNet (nnUNet) architecture was used to segment cardiac chamber volumes and left ventricular (LV) myocardium. We applied these models to patients in the external populations (dark blue) to quantify CAC and cardiac chamber volumes, where we evaluated correlations between measures and associations with outcomes (light blue). CT computed tomography, CV cardiovascular, ECG electrocardiogram, LA left atrium, RA right atrium, RV right ventricle.

**Table 1 | Population characteristics stratified by extent of coronary artery calcification (CAC)**

|  | CAC 0 N = 4618 | CAC 1–100 N = 9006 | CAC 101–400 N = 4816 | CAC > 400 N = 5914 | p-value |
|---|---|---|---|---|---|
| Age | 59 (56–62) | 60 (57–63) | 61 (58 –66) | 63 (59–68) | <0.001 |
| Male | 1681 (36.4%) | 4786 (53.1%) | 3233 (67.1%) | 4741 (80.2%) | <0.001 |
| Pack years smoking | 44 (37.5–60) | 46 (38–63) | 50 (40–70) | 54 (42–75) | <0.001 |
| Past Medical History |  |  |  |  |  |
| Hypertension | 1145 (24.8%) | 2811 (31.2%) | 1817 (37.7%) | 2770 (46.8%) | <0.001 |
| Diabetes | 234 (5.1%) | 675 (7.5%) | 499 (10.4%) | 949 (16.0%) | <0.001 |
| Heart disease | 176 (3.8%) | 475 (5.3%) | 575 (11.9%) | 1931 (32.7%) | <0.001 |
| COPD | 221 (4.8%) | 422 (4.7%) | 268 (5.6%) | 336 (5.7%) | 0.016 |
| Stroke | 84 (1.8%) | 189 (2.1%) | 145 (3.0%) | 263 (4.4%) | <0.001 |
| LV volume (mL) | 92 (79–106) | 99 (84–115) | 104 (89–120) | 110 (95–128) | <0.001 |
| LA volume (mL) | 56 (48–65) | 61 (51–71) | 64 (55–76) | 68 (58–80) | <0.001 |
| RV volume (mL) | 120 (102–142) | 131 (109–156) | 139 (115–163) | 145 (124–168) | <0.001 |
| RA volume (mL) | 73 (63–86) | 79 (67–92) | 83 (71–97) | 88 (75–102) | <0.001 |
| LV mass (g) | 97 (84–114) | 107 (89–127) | 115 (96–134) | 122 (104–142) | <0.001 |
| Shape Index | 0.593 (0.561–0.626) | 0.596 (0.563–0.629) | 0.595 (0.563–0.631) | 0.603 (0.569–0.641) | <0.001 |
| Eccentricity Index | 0.835 (0.815–0.853) | 0.833 (0.812–0.851) | 0.833 (0.812–0.851) | 0.828 (0.807–0.848) | <0.001 |

Categorical variables presented as number(frequency), continuous variables as median (interquartile range).
Categorical variables are compared using Pearson's χ2 test and continuous variables were compared Kruskal–Wallis test, with all tests being two-sided. No adjustment was made for multiple comparisons. *COPD* chronic obstructive pulmonary disease, *LA* left atrial, *LV* left ventricle, *RA* right atrial, *RV* right ventricle.

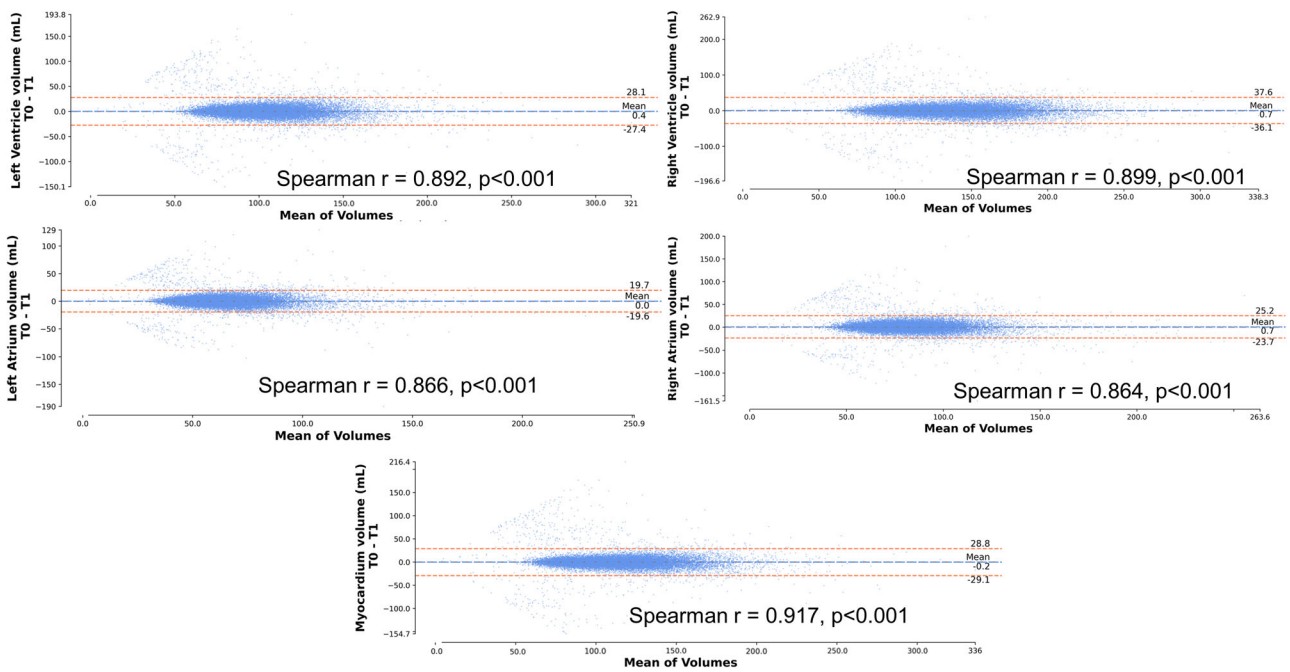

**Fig. 2 | Correlation between baseline and follow-up values.** Correlation between estimates of cardiac volumes and left ventricular mass estimated from computed tomography scans performed at baseline compared to estimated values from scans performed at 1 year in 22292 patients. Each patient shown as a blue dot with limits of agreement as orange dashed lines.

## Associations with Cardiovascular Mortality - NLST

Kaplan-Meier curves for cardiovascular mortality stratified by CAC are shown in Fig. 4, and further detailed in Supplemental Table 1. Quartiles of LV, LA, RV, and RA volume provided risk stratification for cardiovascular mortality as shown in Supplemental Fig. 6. Incidences of cardiovascular mortality in patients with normal compared to abnormal chamber volumes are shown in Supplemental Fig. 7.

Associations with cardiovascular mortality in the multivariable model are shown in Table 2. Patients with CAC 101–400 (adjusted subHR 2.59, 95% CI 1.67–4.03, *p* < 0.001) and CAC > 400 (adjusted subHR 3.57, 95% CI 2.31–5.54, *p* < 0.001) were at significantly increased risk of cardiovascular mortality. Increasing LA volume (adjusted HR 1.14, 95% CI 1.05–1.24, *p* = 0.001), and LV mass index (adjusted HR 1.26, 95% CI 1.05–1.51, *p* = 0.012) were also associated with increased risk of cardiovascular death.

Adjusted associations with cardiovascular mortality were similar when limited to patients without a history of heart disease (Supplemental Table 3). In patients with a history of heart disease CAC was not

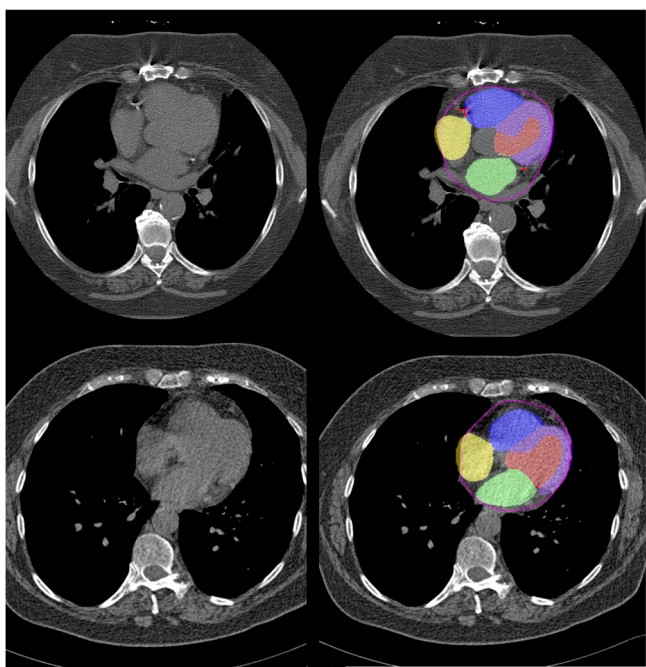

**Fig. 3 | Case examples.** Representative non-contrast computed tomography slices for two patients (left), with super-imposed segmentations (right). One artificial intelligence (AI) model was used to segment a cardiac mask (magenta line) and coronary artery calcium (red). A second AI model segmented left ventricular myocardium (purple), left atrial (green), left ventricle (light red), right ventricle (blue) and right atrial (yellow) volumes.

**Table 2 | Associations with cardiovascular mortality**

| | Adjusted subHR (95% CI) | *p*-value |
|---|---|---|
| CAC 0 | Reference | Reference |
| CAC 1–100 | 1.53 (0.99–2.36) | 0.053 |
| CAC 101–400 | 2.59 (1.67–4.03) | <0.001 |
| CAC > 400 | 3.57 (2.31–5.54) | <0.001 |
| Left ventricular volume (per 10 mL) | 1.12 (1.01–1.24) | 0.040 |
| Right ventricular volume (per 10 mL) | 0.91 (0.85–0.98) | 0.008 |
| Left atrial volume (per 10 mL) | 1.14 (1.05–1.24) | 0.001 |
| Right atrial volume (per 10 mL) | 0.88 (0.81–0.96) | 0.004 |
| LV mass index (per 10 g/m2) | 1.26 (1.05–1.51) | 0.012 |
| Shape index (per 0.1) | 1.45 (0.84–2.51) | 0.179 |
| Eccentricity index (per 0.1) | 1.54 (0.59–4.00) | 0.373 |
| Age (per 10 years) | 1.42 (1.17–1.73) | <0.001 |
| Male | 1.07 (0.82–1.39) | 0.627 |
| Pack years smoking (per 10 years) | 1.04 (1.01–1.07) | 0.021 |
| Hypertension | 1.10 (0.90 –1.34) | 0.369 |
| Diabetes | 1.37 (1.07–1.77) | 0.013 |
| Heart disease | 1.12 (0.89–1.41) | 0.325 |
| COPD | 1.66 (1.18–2.33) | 0.004 |
| Stroke | 1.89 (1.33–2.67) | <0.001 |

Non-cardiac mortality was modeled as a competing risk. *CAC* coronary artery calcification, *CI* confidence interval, *COPD* chronic obstructive pulmonary disease, *subHR* subdistribution hazard ratio.

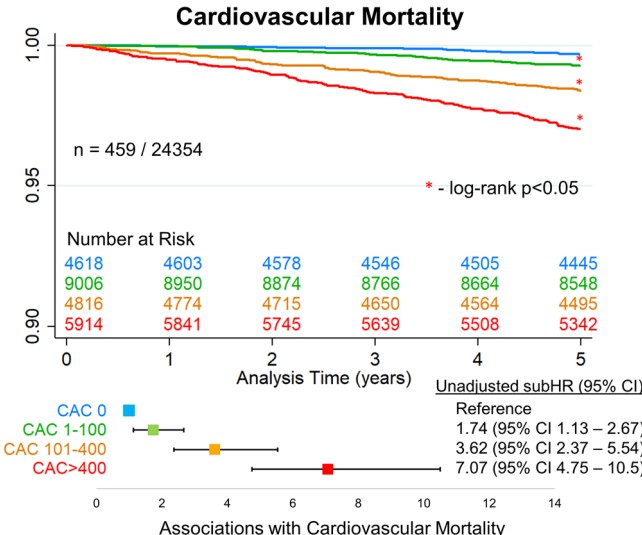

**Fig. 4 | Kaplan-Meier survival curves for cardiovascular mortality.** Kaplan-Meier survival curves for cardiovascular mortality stratified by coronary artery calcium (CAC) scores. The points represent subdistribution hazard ratio (subHR) point estimate, with 95% confidence intervals (CI) shown as whisker plots. Colors reflect CAC score groups as defined in the figure. Source data is available.

cardiovascular abnormalities, all groups of imaging variables significantly improved categorical risk classification, with overall improvement 2.8%–20.3%. However, the combination of all imaging variables led to the greatest improvement, with overall categorical reclassification improvement of 25.9% (95% CI 20.6%–31.2%). Similar results were seen when assessing improvement in classification compared to a multivariable model incorporating age, sex, smoking history, and past medical history.

ROCs for all-cause mortality and cardiovascular mortality are shown in Fig. 5. The AUC for all-cause mortality of the combination of all quantitative imaging variables (AUC 0.657, 95% CI 0.644 – 0.671) was higher than for CAC (AUC 0.638, 95% CI 0.625–0.652), LV mass index (AUC 0.586, 95% CI 0.572–0.600), LA volume (AUC 0.574, 95% CI 0.560–0.588), shape index (AUC 0.538, 95% CI 0.524–0.553, *p* < 0.001), or radiologist identification of abnormalities (AUC 0.523, 95% CI 0.516–0.531, *p* < 0.001). Similarly, AUC for cardiovascular mortality of the combination of all imaging variables (AUC 0.752, 95% CI 0.729–0.775) was higher than for CAC (AUC 0.706, 95% CI 0.683–0.729), LV mass index (AUC 0.674, 95% CI 0.649–0.700), LA volume (AUC 0.633, 95% CI 0.606–0.660), shape index (AUC 0.572, 95% CI 0.544–0.600, *p* < 0.001), or radiologist identification of abnormalities (AUC 0.530, 95% CI 0.514–0.545, *p* < 0.001). Comparison of prediction performance for clinical, imaging, and combined models are shown in Fig. 6.

**EISNER population**

We included 2014 patients who underwent CAC scanning as part of the Early Identification of Subclinical Atherosclerosis by Noninvasive Imaging Research (EISNER) trial to provide external validation in a healthier population. During median follow-up 14.6 years (IQR 1.9–17.4) cardiac death or MI occurred in 74 (3.7%) patients. Population characteristics stratified by occurrence of cardiac death or MI are

associated with cardiovascular death, but LV mass index was (adjusted subHR 1.37 per 10 g/m², 95% CI 1.04–1.80, *p* = 0.025). Associations with cardiovascular mortality in patients without reported cardiovascular abnormalities were similar to the primary analysis (Supplemental Table 4). Results stratified by tube voltage and slice thickness are shown in Supplemental Tables 5 and 6.

Categorical NRI results for cardiovascular mortality are shown in Supplemental Tables 7-8. Compared to radiologist identified

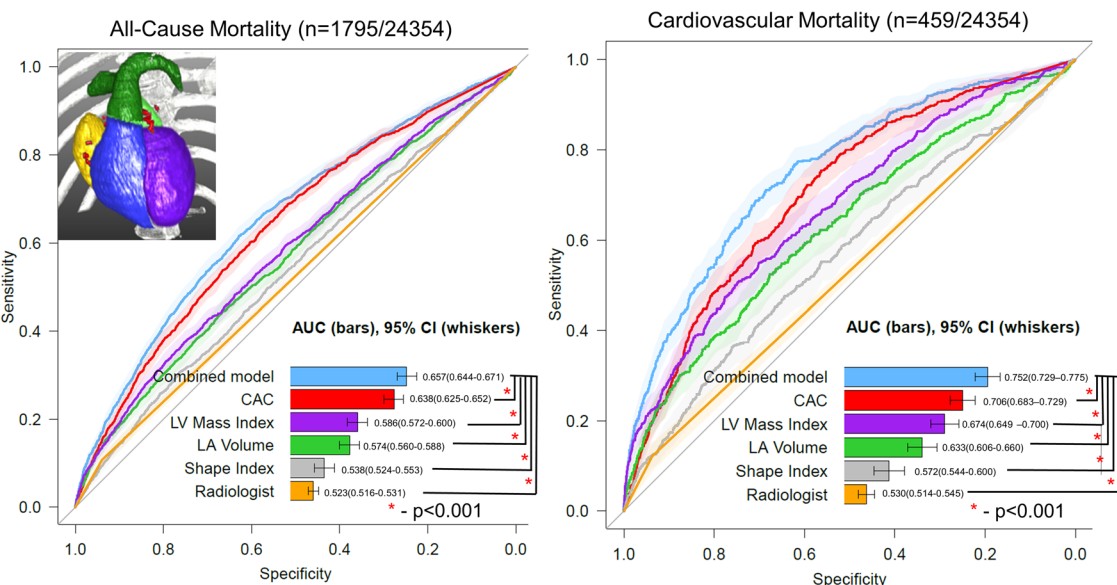

**Fig. 5 | Receiver operating characteristic curves for all-cause mortality and cardiovascular mortality using features.** Receiver operating characteristic curves for all-cause mortality and cardiovascular mortality for coronary artery calcium (CAC) (red), left ventricular (LV) mass index (purple), left atrial (LA) volume (green), shape index (gray) and a combination of all imaging variables (blue). Radiologist identification of abnormality shown in orange. Variables in the combined model were included as continuous variables and integrated using logistic regression analyses. The center of the bar represents area under the receiver operating characteristic curve (AUC), with 95% confidence intervals (CI) shown with whisker plots. AUC is compared using DeLong's test. Source data is available.

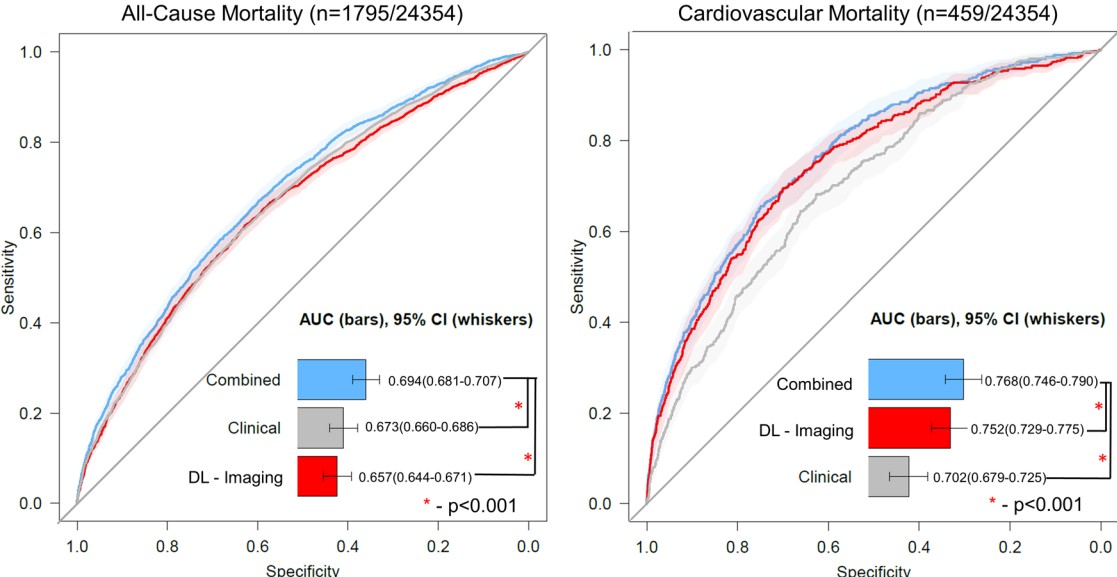

**Fig. 6 | Receiver operating characteristic curves for all-cause mortality and cardiovascular mortality using models.** Receiver operating characteristic curves for all-cause mortality and cardiovascular mortality for clinical (age, sex, medical history - gray), deep learning (DL) derived imaging variables (coronary artery calcium, cardiac volumes, shape index, eccentricity index, and left ventricular mass index – red), and a combined model incorporating all variables (DL derived and clinical variables - blue). Variables in the combined model were integrated using logistic regression analyses. The center of the bar represents area under the receiver operating characteristic curve (AUC), with 95% confidence intervals (CI) shown with whisker plots. AUC is compared using DeLong's test. Source data is available.

shown in Supplemental Table 9. Median LA volume was higher (67.9 ml vs 60.6 ml, $p < 0.001$) and prevalence of CAC > 400 was higher (27.0% vs 6.4%, $p < 0.001$) in patients who experienced cardiac death or MI. Associations with cardiac death or MI are shown in Supplemental Table 10. Models combining clinical and imaging data (AUC 0.804, 95% CI 0.759–0.849, $p < 0.001$) and imaging data alone (AUC 0.792, 95% CI 0.746–0.838, $p = 0.012$) had higher AUC for cardiac death or MI compared to a clinical model incorporating age, sex, and medical history (AUC 0.715, 95% CI 0.653–0.776) as shown in Supplemental Fig. 8.

Including DL-imaging features also significantly improved categorical and continuous NRI (Supplemental Table 11). Risk stratification for cardiac death or MI in young (age <60 years) non-smokers is shown in Supplemental Table 12.

### Low-dose CT population
We included 3319 patients referred for myocardial perfusion imaging who underwent low-dose, ungated CT for attenuation correction of the perfusion scan to provide further external validation. During

median follow-up 2.9 years (IQR 1.6–5.0) death or MI occurred in 177 (5.3%) patients. Population characteristics stratified by occurrence of cardiac death or MI are shown in Supplemental Table 13. Median LA volume (85.0 ml vs 76.4 ml, $p < 0.001$) and median LV volume (129.4 ml vs 118.4 ml, $p < 0.001$) were higher in patients who experienced death or MI. Associations with death or MI are shown in Supplemental Table 14. Receiver operating characteristic curves for death or MI using clinical, imaging, and combined models are shown in Supplemental Fig. 9. Including DL-imaging features also significantly improved categorical and continuous NRI (Supplemental Table 15).

## Discussion

We evaluated whether quantifying CAC, cardiac volumes, LV mass, and ventricular morphology using two previously validated AI models could improve risk stratification of patients undergoing non-cardiac lung CT scans in a large external population from the NLST trial. We demonstrated that LV mass index and LA volume from non-contrast, ungated CT scans are associated with all-cause and cardiovascular mortality. We also found that higher deep learning (DL)-derived CAC was associated with an increased risk of both all-cause mortality and cardiovascular mortality. Importantly, we also demonstrated that a combined model incorporating CAC, cardiac volumes, LV mass index, and ventricular morphology had higher prediction performance than any measure in isolation. Furthermore, the combined model improved categorical risk classification of over 25% of patients compared to the current clinical standard, radiologist identification of cardiovascular abnormality. We went on to show associations between CAC, cardiac volumes, LV mass index and clinical outcomes in two additional external populations with different risk profiles. Given that in the United States, 428 CT scans are performed per 1000 adults each year[25], this approach could be used to improve identification of cardiovascular abnormalities and estimation of cardiovascular risk for a substantial number of patients.

Chest imaging is one of the most frequently performed CT examinations[15], with dedicated cardiac imaging representing a small fraction of those. While incidental cardiac findings may be seen on over half of those studies, it is only reported on 3-31% of studies[26]. It is possible that Radiologists could identify additional abnormalities if specifically focused on cardiac incidentals, but this is not the case in typical clinical practice. This is consistent with our finding that only 14% of patients with CAC > 400 and less than 10% of patients with abnormal LV mass index had abnormalities reported. This care gap exists despite guidelines suggesting that CAC be routinely assessed on all non-cardiac chest CT scans[16]. The proposed approach could potentially simplify this process by providing automated estimates of chamber volumes, LV mass index, and CAC for radiologists to incorporate during reporting, leading to substantially improved prediction of cardiovascular mortality and improved risk categorization in over 25% of patients. For example, AI-based identification of CAC (with radiologist oversight) improves adoption of medical therapy when coupled with automated notifications[27]. The CAC model is computationally efficient, providing results in ~6 s (faster than a standard U-Net model)[28]. We paired this model with a recently developed AI model for automated segmentation of structures from CT, with all results available in ~18 s. The combined workflow is fully automated and therefore could be readily incorporated into most clinical workflows without significant disruptions.

Our study demonstrates that cardiac chambers and LV myocardium can be estimated from non-contrast chest CT to improve risk classification. Patients with higher atrial or ventricular volumes and abnormal LV mass index were more likely to experience all-cause or cardiovascular mortality. Additionally, left atrial volume and LV mass index were associated with increased risk of both all-cause and cardiovascular mortality after adjusting for relevant confounding factors and all other imaging variables. We applied thresholds for abnormal

cardiac volumes, which were based on a study of healthy individuals undergoing cardiac CT. While we did identify significant associations with cardiac outcomes, our results highlight the need for age and sex-specific normal values. While previous studies have demonstrated that CT-derived left ventricular volumes[13] and left ventricular hypertrophy[29] are associated with adverse cardiovascular events, these studies were performed using contrast-enhanced studies. It may be possible for radiologists to provide chamber volume estimates from non-contrast scans, but this is not currently routinely performed and would likely have high inter-reader variability and be too time-consuming for routine clinical use. Lastly, we evaluated shape index and eccentricity index, which are measures of ventricular morphology. We demonstrated that higher shape index, representing a more spherical LV cavity, was independently associated with both all-cause and cardiovascular mortality. Similar volume measurements (but not CAC) can also be performed using cardiovascular magnetic resonance[30], but capacity is limited at most centers and our proposed DL-based estimates can be performed on any non-contrast CT. Importantly, associations were similar in patients without reported cardiac abnormalities. Left atrial volume and LV mass index were associated with cardiovascular mortality in patients with a history of cardiac disease (but not CAC, potentially due to inclusion of patients with previous stents or bypass grafts), providing a valuable method for risk stratification in this population. Similar results were demonstrated in a younger population of patients undergoing CAC scanning as part of the EISNER trial as well as in a third external cohort of patients undergoing low-dose, ungated CT with myocardial perfusion imaging. Lastly, we demonstrated that incorporating all of the imaging variables had the highest prediction performance for all-cause and cardiovascular mortality while also leading to the greatest improvement in categorical risk classification compared to radiologist identification of abnormalities.

In three large, external cohorts we demonstrated that CAC scores obtained in a fully automated manner using DL were associated with all-cause and cardiovascular mortality. Chiles et al. demonstrated that expert physician evaluation of CAC, with formal scoring or estimates, were associated with cardiovascular death with HRs of 6.10 – 6.95 for the highest CAC groups in a group of 1575 patients[31]. These risks are similar to that seen for CAC > 400 in our analysis in a much larger sample. Zeleznik et al. developed a U-net based DL model which automatically quantified CAC and showed that in a subset of 14959 patients from the NLST, subjects with CAC > 400 had an unadjusted HR 5.98 compared to CAC 0 for cardiovascular death[20]. The higher risk demonstrated in our study (unadjusted subHR 7.07) could be explained by improved classification of patients with CAC 0 which are used as the reference risk group. We previously demonstrated favorable results for the cLSTM model compared to a U-net based model[28]. Additionally, it is notable that the associations in our study were also present between CAC and all-cause mortality, suggesting that targeted interventions could potentially influence overall survival.

Our study has a few important limitations. We evaluated associations with cardiovascular death, but it is possible the cause of death is misclassified in some patients. However, results were similar when looking at associations with all-cause death. We have limited information regarding the exact nature of cardiovascular abnormalities which were identified. Therefore, we are not able to determine how frequently the identified abnormality was significant coronary calcification compared to other identifiable abnormalities, such as chamber enlargement or valve calcification. Similarly, we do not have further classification of history of heart disease. Additionally, we do not know if physicians initiated medical therapy in response to CT findings, which may decrease the associations between imaging findings and outcomes. While we did not assess the correlation between DL measurements and expert segmentations of CAC, we have previously demonstrated excellent intraclass correlation between the DL and

expert CAC measurements[22,28]. We performed several analyses in all three populations and some associations may be related to chance alone. However, the associations with DL-based imaging features were consistent across analyses and the likelihood of making multiple type 1 errors for the same variable would be minimal and applying corrections for multiple testing can increase the rate of type 2 error[32]. We did not incorporate race or ethnicity into our analyses. The majority of patients in the NLST trial were white (91% in our cohort); future studies should evaluate methods to incorporate more diverse populations. Lastly, we utilized DL models to extract known anatomic features; using DL to directly predict outcomes may lead to identification of latent features associated with outcomes. However, explanatory mechanisms would need to be implemented and validated in such an approach to warrant clinical use.

Imaging biomarkers—CAC, cardiac chamber volumes, LV mass, and shape index—can be automatically and rapidly quantified using DL from non-cardiac CT scans. These estimates were predictive of all-cause and cardiovascular mortality. DL-derived CAC scores improved classification of patients compared to expert identification of cardiovascular abnormalities. Routine measurements of all these parameters can potentially enhance risk stratification and improve clinical decision-making in the management of patients at risk of cardiovascular disease.

## Methods

### Study populations

The overall design of this retrospective study is shown in Fig. 1. This study utilized de-identified images from 3 separate external testing cohorts including patients from the NLST (NCT00047385), a multi-center randomized controlled trial of patients randomized to low-dose chest CT for lung cancer screening[33], asymptomatic patients from the EISNER trial (NCT00927693) who underwent CAC scanning[34], and patients from two centers who underwent myocardial perfusion imaging with low-dose, ungated, chest CT for attenuation correction. The NLST trial included current or former heavy smokers between the ages of 55 and 74. Patients were randomly assigned to non-contrast, non-ECG-gated chest CT imaging between 2002 and 2007. We included 24805 subjects from this external cohort (previously unseen by the DL models) with available baseline CT imaging and follow-up for mortality. Of those, we excluded cases where image files were corrupt (n = 133, 0.5%), the scan length was less than 12 cm or did not include the heart (n = 292, 1.2%), and cases where segmentation failed (n = 26, 0.1%), leaving 24354 subjects. In cases where segmentation failed, neither the CAC nor the cardiac volume model was able to process the scan. The baseline CT was used to assess associations with outcomes. We also compared estimates from baseline CT scans with estimates from CT scans performed at 1 year in 22292 patients as a measure of stability. Lastly, we compared cardiac volumes and left ventricular mass in a cohort of 80 patients from a clinical trial. These patients underwent low-dose, ungated CT and contrast-enhanced, ECG-gated, cardiac CT angiography on the same day, during a single imaging session, minimizing potential differences between scans (NCT02110303). Cardiac volumes and mass from ECG-gated, contrast-enhanced CT were annotated manually by experienced clinicians using dedicated software (Syngo.Via, Siemens Healthineers, Erlangen, Germany). The study protocol complied with the Declaration of Helsinki and was approved by the institutional review boards at participating institutions. The study used de-identified image sets and did not collect new data, therefore the research is considered non-human subject research.

### Clinical data

For the NLST population, past medical history and smoking history were collected during the course of the trial[33]. Additionally, clinical interpretation from each scan was recorded as part of the NLST trial including the presence or absence of clinically relevant cardiovascular abnormalities such as CAC or cardiomegaly. Patients had follow-up for all-cause mortality and information from death certificates regarding underlying cause. Cardiovascular mortality was determined for the ICD-10 codes using established definitions for cardiovascular mortality[35], and validated ICD-10 codes[36]. For the EISNER population, medical history was determined at baseline and patients were followed prospectively for occurrence of cardiovascular death or myocardial infarction[34]. For the third external population, demographics and medical history were determined at the time of CT scanning and incidence of all-cause mortality of myocardial infarction was determined from administrative databases.

### CT image acquisition and reconstruction

Images were acquired at each participating site with site-specific protocols. Anonymized image datasets were used for the present analysis. In the NLST population, CT scans were acquired using 17 different camera systems, including systems manufactured by GE Healthcare, Philips, Siemens, and Toshiba. Most patients were imaged with tube voltage of 120 kVp (n = 21287, 87.5%) followed by 140 kVp (2313, 10.5%), and a small number of patients were imaged with other tube voltages (80 kVp n = 377, 90 kVp n = 141, 100 kVp n = 21, 130 kVp n = 4, 135 kVp n = 211). Median tube current was 60 mA (interquartile range 45–72). Mean pixel size was 0.67 mm and ranged from 0.44–0.98 mm. Reconstruction thickness ranged from 1–6 mm, with most patients having slice thickness of 2.5 mm (46.9%) or 2 mm (30.1%). For the EISNER population, patients underwent standard CAC scans including a single scan of ~30–40 slices which were 3 mm or 2.5 mm in thickness. For the remaining external cohort, CT scans were performed with a helical acquisition with a tube voltage of 120 kVp and slice thickness of 3.0 mm (n = 1878) or 5.0 mm (n = 1441).

### Model architectures and training

We utilized our previously validated DL model for CAC segmentation[28]. In brief, the system consists of two networks, the first of which is trained for segmentation of the heart silhouette and the second network was trained to segment the CAC. A supervised learning regimen was used for both segmentation networks. The heart mask was applied to the final CAC prediction to reduce bone overcalling or calcification in non-cardiac regions. For training, internal validation, and internal testing, we used data from 3 centers that included 9543 scans (1827 ECG-gated CAC scans and 7716 CT attenuation scans)[22,23]. The model includes a correction factor for slice thickness, ensuring consistent scoring in spite of differences in slice thickness. CAC scores are automatically obtained from the DL segmentations using established methods[3].

Cardiac chamber volumes and LV myocardium were segmented using TotalSegmentator[21]. The model utilizes the no new-net UNet (nnU-Net) architecture[37] to automatically segment a variety of anatomic structures from images. Expert annotations from contrast images were transferred to registered non-contrast images sets to train the model to segment the same structures on non-contrast image sets. During model validation, the Dice score for LV myocardium, LV, left atrium (LA), right ventricle (RV) and right atrium (RA) ranged from 0.95–0.97[21]. Three-dimensional segmentations for one patient are shown in Supplemental Figure 10. LV myocardial volume was used to calculate LV mass, using a density factor of 1.055[38]. Abnormal LV mass was defined as volume >97.5th percentile using sex-specific normal limits indexed to body surface area (>80 g/m$^2$ for men, >65 g/m$^2$ for women)[39]. Similarly, we defined abnormal cardiac volumes for women (LV volume > 147 mL, RV volume > 180 mL, LA volume > 99 mL and RA volume >126 mL) and men (LV volume > 195 mL, RV volume > 240 mL, LA volume > 121 mL and RA volume > 162 mL) based on >97.5th percentile of normal volumes using sex-specific normal limits[39]. A comparison of patient classifications at baseline compared to follow-up

imaging at 1 year is shown in Supplemental Table 16. Lastly, we quantified the major axis (length) and the two minor axis measurements for the LV volume segmentations. Shape index was calculated as the ratio of the maximal minor axis dimension to the major axis dimension, similar to the method applied in myocardial perfusion imaging[40]. Eccentricity index was calculated as: 1-(minor axis*minor axis/length$^2$). Lower values for shape index signify relative elongation of the LV, while higher values are seen in more spherical remodeling patterns.

### Statistical analysis

Continuous variables were summarized as mean (standard deviation [SD]) if normally distributed and compared using a Student's t-test. Continuous variables that were not normally distributed were summarized as median (interquartile range [IQR]) and compared using a Mann-Whitney U-test or Kruskal–Wallis test. Agreement between estimates from low-dose, ungated CT and measurements from contrast-enhanced, ECG-gated, cardiac CT scans, and agreement between estimates from baseline and 1 year CT scans was assessed with Spearman's correlation and visualized with Bland-Altman plots.

Associations with all-cause mortality were assessed with univariable and multivariable Cox proportional hazards analyses. The multivariable model included the variables of interest (LA, LV, RA, and RV volume, LV mass, shape index, eccentricity index, and CAC) as well as potential confounders including age[41], sex[41], history of COPD[42], diabetes[43], hypertension[44], heart disease[45], and stroke[46] as well as smoking history[47]. The suspected relationships between these variables are outlined in Supplemental Figure 11. Associations with cardiovascular mortality were evaluated with Fine-Gray competing risk analyses, with non-cardiovascular mortality as a competing risk. For the EISNER and low-dose CT populations there were insufficient events to simultaneously evaluate all variables, so multivariable models were created using stepwise backward elimination. In the NLST population, associations were separately assessed in patients with and without a history of heart disease as well as in patients without radiologist identified cardiovascular abnormalities. Lastly, we evaluated for differences in the associations with clinical outcomes according to tube voltage and slice thickness. These analyses were limited to unadjusted analyses due to a low number of events in some groups, resulting in wide confidence intervals. However, we also assessed for differences in associations between tube voltage and slice thickness categories using interaction analyses. The proportional hazards assumption was evaluated with Schoenfeld residuals[48], and found to be valid in all analyses.

In the NLST population, we evaluated prediction performance, using area under the receiver operating characteristic curve (AUC), for all-cause mortality and cardiovascular mortality of CAC, cardiac volume, shape index and a combination of the three measures (from regression models including log CAC as a continuous measure, abnormal cardiac volume as a categorical variable, and shape index as a continuous variable). We also evaluated AUC for a clinical model (age, sex, smoking history and medical history [hypertension, diabetes, heart disease, COPD, stroke]), an imaging model with DL-derived variables (CAC, cardiac volumes, shape and eccentricity index, and LV mass), and a combined model incorporating all variables. Variables were combined using logistic regression. Categorical net reclassification index (NRI) was used to assess the additive prognostic utility of DL CAC, cardiac volumes, shape index, and the combined model[49]. NRI was calculated when added to either radiologist identification of cardiovascular abnormality or all other components of the multivariable model including age, sex, smoking history, and past medical history.

All statistical tests were two-sided, and a p-value < 0.05 was considered statistically significant. All analyses were performed using Stata/IC version 13.1 (StataCorp, College Station, Texas, USA) and R (version 4.1.2) including the "DAGitty" package[50].

### Reporting summary

Further information on research design is available in the Nature Portfolio Reporting Summary linked to this article.

## Data availability

All derived data supporting the findings of this study are available within the paper, in the supplementary information file, and in the source data file. Original data from the NLST can be requested through the National Cancer Institute. Restricted access for the deidentified EISNER, and low-dose CT populations can be obtained via requests to the corresponding author Dr. Piotr Slomka (Piotr.Slomka@cshs.org). Requests should include the name and contact details of the person requesting the data, which data and clinical variables are requested and the purpose of requesting the data. Requests will be subject to consideration by the steering committees of the cohorts and the investigational review board of Cedars-Sinai Medical Center and investigational review boards from other centers if applicable. Time frame for a response will be within 3 months. Data requests under agreement will be considered for the purpose of reproducing the data and subject to appropriate confidentiality obligations and restrictions. Source data are provided with this paper.

## Code availability

The TotalSegmentator code is publicly available[21] at https://github.com/wasserth/TotalSegmentator, and the cLSTM code is available at https://doi.org/10.5281/zenodo.10632288[51].

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

## Acknowledgements

This research was supported in part by grant R35HL161195 from the National Heart, Lung, and Blood Institute/ National Institutes of Health (NHLBI/NIH) (PI: PS) as well as RO1EB034586 from the National Institute of Biomedical Imaging and Bioengineering (PI: PS). The authors thank the National Cancer Institute for access to NCI's data collected by the National Lung Screening Trial (NLST) accessed under project number NLST-981. The content is solely the responsibility of the authors and does not necessarily represent the official views of the National Institutes of Health.

## Author contributions

R.J.H.M. participated in study design, data analysis, manuscript drafting and manuscript revisions. A.K. participated in study design, data analysis, and manuscript revisions. A.S. participated in study design, data analysis, and manuscript revisions. B.B. participated in study design, data analysis, and manuscript revisions. A.M.M. participated in data analysis, and manuscript revisions. T.D.R. participated in study design, data collection, and manuscript revisions. A.J.E. participated in study design, data collection, and manuscript revisions. D.E.N. participated in study design, data collection, and manuscript revisions. M.L. participated in data analysis, and manuscript revisions. K.P. participated in study design, data analysis, and manuscript revisions. S.D.V.K. participated in data analysis and manuscript revisions. P.B.K. participated in data analysis, and manuscript revisions. J.X.L. participated in study design and manuscript revisions. C.H. participated in study design and manuscript revisions. D.D. participated in study design, and manuscript revisions. D.S.B participated in study design and manuscript revisions. P.J.S. participated in study design, data analysis, and manuscript revisions.

## Competing interests

R.J.H.M. has received consulting fees and research support from Pfizer. D.S.B. and Slomka and P.B.K. participate in software royalties for QPS software at Cedars-Sinai Medical Center. P.J.S. has received research grant support from Siemens Medical Systems and has served as a consultant for Synektik. D.S.B. has served as a consultant for GE Healthcare. A.M.M. is supported by a research scholarship from the Polish National Agency for Academic Exchange. A.J.E. reports receiving a speaker's fee from Ionetix, consulting fees from W. L. Gore & Associates, authorship fees from Wolters Kluwer Healthcare—UpToDate, and serving on a scientific advisory board for Canon Medical Systems USA; his institution has grants/grants pending from Attralus, Bruker, Canon Medical Systems USA, Eidos Therapeutics, GE HealthCare, Intellia Therapeutics, Ionis Pharmaceuticals, Neovasc, Pfizer, Roche Medical Systems, and W. L. Gore & Associates. T.D.R. has received research grant support from GE HealthCare and Pfizer. D.E.N. reports a grant from the British Heart Foundation and Wellcome Trust. The remaining authors declare no competing interests.
