## [Peer Review File · Nature Communications]

REVIEWER COMMENTS

Reviewer #1 (Remarks to the Author):

The authors made extensive revisions and improved the manuscript. Yet, the manuscript still has major flaws that need to be addressed. The remaining main weakness lies within the limited potential clinical applicability and limited evidence of validity that the measures are actually accurate on a patient level.

1. Major flaws

a. While the additionally introduced cohorts may be used to improve the manuscript significantly, the authors currently fail to conduct the most relevant investigations. The main goal of the manuscript is to show improvements in risk stratification by the inclusion of volumetric measures in combination with CAC using a fully automatic approach. Yet, the currently done additional analyses do not help to investigate this knowledge gap. The presented results of known risk factors are logical and not surprising, which is good, but the authors need to go the necessary step further and investigate if the improvement in risk stratification is also met in these cohorts. This means they need to do the analyses of the main manuscript (as done for NLST) also in these two newly introduced cohorts and show actual improvements in risk stratification by using their model as compared with the clinical standard and a simple model. Also, as discussed by reviewer #2, the clinical model should include known clinical factors. Further, clinical application needs definite cut-offs – this means the authors should use the cutoff values from NLST and apply those to the two independent cohorts; then they should perform the same analyses as done for NLST, individually in each cohort. This would add the necessary evidence, that actual meaningful cut-off values exist and that they can be transferred and used across cohorts. The sole usage of quantiles in each cohort is insufficient.

b. The statistical analysis also needs improvement. Are the p-values of the association analyses of the two additional cohorts adjusted for multiple comparisons? This would be needed. Further, the stepwise elimination is not sufficient and lacks adjustment for known and clinically highly relevant confounders, such as sex, hypertension, or dyslipidemia; also the adjustments should use the same approach across all three cohorts to improve interpretability and readability. Also, it needs to be explicitly shown that the model works in stratified analyses for young participants and non-smokers.

c. The Bland Altman Plots of Figure 2 are a great way to show the analyses of baseline versus one-year follow-up. Yet, the results seem to need more detailed further analyses. There is a clear non-random distribution, especially showing way greater variation in smaller mean volumes < 100 mL. In this subgroup, it seems that there are variations of more than +/- 100% between T0 and T1. As discussed

earlier, the actual cut-off values of the quantiles need to be described and applied across all cohorts, this is especially also needed to help interpret the Bland Altman Results, such as it needs to be shown how many patients change risk groups between T0 and T1 and to what extent. It needs to be investigated how many patients with low scores may show a more than doubled volume in T1 and vice versa. This has a huge clinical impact and could lead to major misclassifications and potentially harm rather than benefit a patient in clinical practice.

d. The authors did change the publishing license. This is good. Also, the model should be uploaded to an open-access repository and not only be available on reasonable request as this limits reproducibility.

e. The granularity of radiological heart assessments of NLST should be shown.

Reviewer #2 (Remarks to the Author):

I commend the authors for doing an excellent and thorough job in responding to the input of both Reviewers. I appreciate that the authors have adequately addressed the concerns.

Regarding clarification of the results relative to already available clinical variables, because CAC and structural results are so confounded by other strong predictors of mortality and CV outcomes, especially age, sex, smoking, diabetes,... that there seems to be little value in presenting/emphasizing non-risk-adjusted results like in Figures 4 and 5 (and Supplemental Figures 4 & 5). Similarly, I think the predictive value of the models evaluated here could be put in better clinical perspective by including in manuscript Figure 6 (rather than Supplemental Figure 6) the ROC for clinical models.

We would like to thank the Reviewers for their careful review of our manuscript. We have made further revisions to the manuscript in response to the additional Reviewers' suggestions. As a result, we believe the manuscript is significantly improved and hope it is now judged acceptable for publication. We have copied reviewers' comments verbatim below in bold font. Our point-by-point response is in normal font, with new or revised portions of the text in red.

Reviewer #1 (Remarks to the Author):

The authors made extensive revisions and improved the manuscript.

Thank you for the time and effort spent reviewing our manuscript.

1. Yet, the manuscript still has major flaws that need to be addressed. The remaining main weakness lies within the limited potential clinical applicability and limited evidence of validity that the measures are actually accurate on a patient level.

Thank you for this helpful critique. In the previous revision, we included a paired analysis between ungated non-contrast CT and gated contrast-enhanced CT. This analysis demonstrated good agreement between values (correlations: LV myocardium $r=0.947$, LA volume $r=0.926$, RA volume $r=0.893$, LV volume $r=0.793$, and RV volume $r=0.922$). However, we are not able to perform a similar paired analysis with the NLST dataset, as only non-contrast data are available in those patients. We do agree with you that it is important to evaluate patient classification based on thresholds for abnormal values. We have performed the additional analyses you have suggested below which we believe also highlight the ability of the model to provide consistent patient classifications.

Major flaws

2. While the additionally introduced cohorts may be used to improve the manuscript significantly, the authors currently fail to conduct the most relevant investigations. The main goal of the manuscript is to show improvements in risk stratification by the inclusion of volumetric measures in combination with CAC using a fully automatic approach. Yet, the currently done additional analyses do not help to investigate this knowledge gap. The presented results of known risk factors are logical and not surprising, which is good, but the authors need to go the necessary step further and investigate if the improvement in risk stratification is also met in these cohorts. This means they need to do the analyses of the main manuscript (as done for NLST) also in these two newly introduced cohorts and show actual improvements in risk stratification by using their model as compared with the clinical standard and a simple model. Also, as discussed by reviewer #2, the clinical model should include known clinical factors.

Thank you for this helpful suggestion. We have now performed comparisons for prediction performance between clinical models, imaging models, and combined models utilizing the new information derived from deep learning. The clinical models included age, sex, and medical history (hypertension, diabetes, dyslipidemia, family history, and smoking). The imaging models included all DL-based imaging parameters (coronary artery calcium, cardiac volumes, shape, eccentricity index, and left ventricular mass index). In the EISNER population, the combined imaging model (AUC 0.804, 95% CI 0.759 – 0.848, $p < 0.001$) and imaging model (AUC 0.792, 95% CI 0.746 – 0.838, $p = 0.012$) had higher AUC compared to the clinical model (AUC 0.715, 95% CI 0.653 – 0.776). In the low-dose CT population, the combined model (AUC 0.704, 95% CI 0.668 – 0.741) had higher AUC compared to the imaging model (AUC 0.668, 95% CI 0.628 – 0.707, $p = 0.004$) and the clinical only model (AUC 0.643, 95% CI 0.603 – 0.682). These results suggest that incorporating imaging and clinical features can be used to improve risk stratification.

We also went on to evaluate the categorical and continuous net reclassification index (NRI) for the combined models compared to the clinical-only model (testing the incremental benefit of adding the imaging parameters). In both populations, the combined model significantly improved continuous and categorical NRI. The relevant changes to the manuscript include:

In Results:

Models combining clinical and imaging data (AUC 0.804, 95% CI 0.759 – 0.848, $p < 0.001$) and imaging data alone (AUC 0.792, 95% CI 0.746 – 0.838, $p = 0.012$) had higher AUC for cardiac death or MI compared to a clinical model incorporating age, sex, and medical history (AUC 0.715, 95% CI 0.653 – 0.776) as shown in **Supplemental Figure 8**. Including DL-imaging features also significantly improved categorical and continuous NRI (**Supplemental Table 11**).

Also in Results:

Receiver operating characteristic curves for death or MI using clinical, imaging, and combined models are shown in **Supplemental Figure 9**. Including DL-imaging features also significantly improved categorical and continuous NRI (**Supplemental Table 15**).

In Supplement:

Supplemental Figure 8

Cardiac death or myocardial infarction N=2014

Supplemental Figure 8: Receiver operating characteristic curves for cardiac (CV) death or myocardial infarction (MI) for the Early Identification of Subclinical Atherosclerosis by Noninvasive Imaging Research trial. The models included were clinical (age, sex, medical history), deep learning (DL) derived imaging variables (coronary artery calcium, cardiac volumes, shape index, eccentricity index, and left ventricular mass index), and a combined model incorporating all variables (DL derived and clinical variables). Variables were integrated using logistic regression analyses. AUC – area under the receiver operating characteristic curve, CI – confidence interval

Supplemental Figure 9

All-cause mortality or myocardial infarction N=3319

Supplemental Figure 9: Receiver operating characteristic curves for death or myocardial infarction (MI) in the low dose computed tomography population. The models included were clinical (age, sex, medical history), deep learning (DL) derived imaging variables (coronary artery calcium, cardiac volumes, shape index, eccentricity index, and left ventricular mass index), and a combined model incorporating all variables (DL derived and clinical variables). Variables were integrated using logistic regression analyses. AUC – area under the receiver operating characteristic curve, CI – confidence interval

Supplemental Table 11:

	Low ($<0.25\%/year$)	Intermediate ($0.25-0.5\%/year$)	High ($\geq 0.5\%/year$)		
Without Events					
Low ($<0.25\%/year$)	1101	87	48	Decreased Risk	0.178
Intermediate ($0.25-0.5\%/year$)	239	135	111	Increased Risk	0.127
High ($\geq 0.5\%/year$)	46	61	112	Net Reclassification	0.052
With Events					
Low ($<0.25\%/year$)	242	159	2	Decreased Risk	0.135
Intermediate ($0.25-0.5\%/year$)	18	36	2	Increased Risk	0.270
High ($\geq 0.5\%/year$)	0	0	0	Net Reclassification	0.135
Overall Reclassification (95% CI)				0.187 (0.043 – 0.331)	
Continuous Reclassification (95% CI)				0.778 (0.568 – 0.987)	

Supplemental Table 11: Net-reclassification analysis for cardiac death or myocardial infarction. The predicted risk from clinical data (rows) is compared to the risk from a model combining clinical and imaging data (columns). The clinical model included age, sex, and medical history. The combined model included clinical data as well as deep learning-derived imaging variables (coronary artery calcium, cardiac volumes, shape index, eccentricity index, and left ventricular mass index). CI – confidence interval.

Supplemental Table 15:

	Low ($<1\%/year$)	Intermediate ($1-2\%/year$)	High ($\geq 2\%/year$)		
Without Events					
Low ($<1\%/year$)	595	100	13	Decreased Risk	0.260
Intermediate ($1-2\%/year$)	515	832	256	Increased Risk	0.117
High ($\geq 2\%/year$)	24	279	528	Net Reclassification	0.143

	Low (<1%/year)	Intermediate (1-2%/year)	High (≥2%/year)		
With Events					
Low (<1%/year)	12	5	3	Decreased Risk	0.144
Intermediate (1-2%/year)	9	41	33	Increased Risk	0.228
High (≥2%/year)	0	17	60	Net Reclassification	0.083
Overall Reclassification (95% CI)				0.211 (0.120 – 0.301)	
Continuous Reclassification (95% CI)				0.334 (0.184 – 0.485)	

Supplemental Table 15: Net-reclassification analysis for death or myocardial infarction. The predicted risk from clinical data (rows) are compared to the risk from a model combining clinical and imaging data (columns). The clinical model included age, sex, and medical history. The combined model included clinical data as well as deep learning derived imaging variables (coronary artery calcium, cardiac volumes, shape index, eccentricity index, and left ventricular mass index). CI – confidence interval.

3. Further, clinical application needs definite cut-offs – this means the authors should use the cutoff values from NLST and apply those to the two independent cohorts; then they should perform the same analyses as done for NLST, individually in each cohort. This would add the necessary evidence, that actual meaningful cut-off values exist and that they can be transferred and used across cohorts. The sole usage of quantiles in each cohort is insufficient.

Thank you for this suggestion. We agree that it is important to be able to categorize patients as having normal or abnormal values based on the estimated chamber volumes and left ventricular mass. Since there are established thresholds for normal heart chamber sizes from high-resolution gated cardiac CT angiography with contrast in normal healthy subjects, where all the chambers are precisely delineated, we have applied these previously established thresholds to our datasets. This approach avoids the bias inherent in establishing thresholds within the population that they are tested, allowing us to evaluate this clinical threshold in the NLST population rigorously. Importantly, it also avoids deriving thresholds from a population of older, smokers. Therefore, we utilized the 97.5th percentile of cardiac volumes in normal patients (established by Fuchs et al.) as the threshold for abnormal values. The study by Fuchs et al. included 569 healthy individuals who underwent cardiac CT angiography as part of the Copenhagen General Population study. The mean age was 55 and 67% of patients were

female. Using these thresholds for abnormal volume, we demonstrate that patients with abnormal values cardiac volumes had an increased risk of cardiovascular mortality in the NLST population. However, the number of patients with events as well as the number of patients classified as having abnormal volumes, was limited in the EISNER and low-dose CT (more healthy populations), and consequently, these cohorts are underpowered for such analysis. Importantly, since we did not derive the thresholds in the NLST population (but utilized previously established thresholds) we are providing independent, external validation of the values. The relevant changes to the manuscript include:

In Results:

Incidences of cardiovascular mortality in patients with normal compared to abnormal chamber volumes are shown in **Supplemental Figure 7**.

In Discussion:

We applied thresholds for abnormal cardiac volumes, which were based on a study of healthy individuals undergoing cardiac CT. While we did identify significant associations with cardiac outcomes, our results highlight the need for age and sex-specific normal values.

Methods:

Similarly, we defined abnormal cardiac volumes for women (LV volume > 147mL, RV volume > 180mL, LA volume > 99mL and RA volume >126mL) and men (LV volume > 195mL, RV volume > 240mL, LA volume > 121mL and RA volume >162mL) based on >97.5th percentile of normal volumes using sex-specific normal limits³⁹.

In Supplement:

Supplemental Figure 7:

Kaplan-Meier survival curves for cardiovascular mortality by normal or abnormal chamber volume. In women, the thresholds for abnormal chamber volumes were: left ventricle volume > 147mL, right ventricle volume > 180mL, left atrium volume > 99mL and right atrium volume > 126mL. In men, the thresholds for abnormal chamber volumes were: left ventricle volume > 195mL, right ventricle volume > 240mL, left atrium volume > 121mL and right atrium volume > 162mL. CI – confidence interval

4. The statistical analysis also needs improvement. Are the p-values of the association analyses of the two additional cohorts adjusted for multiple comparisons? This would be needed.

Thank you for helping us clarify this aspect of the analysis. The associations presented in the additional cohorts are not adjusted for multiple comparisons. However, the associations between continuous variables and outcomes are consistent between cohorts. Therefore, the chance of making multiple type 1 errors (for the same variables) would be minimal. Additionally, adjusting for multiple comparisons (with Bonferroni correction for example) is known to increase the type 2 error (reference 35). We have outlined these considerations in the manuscript as follows:

In Limitations:

We performed several analyses in all three populations and some associations may be related to chance alone. However, the associations with DL-based imaging features were consistent across analyses and the likelihood of making multiple type 1 errors for the same variable would be minimal and applying corrections for multiple testing can increase the rate of type 2 error³².

5. Further, the stepwise elimination is not sufficient and lacks adjustment for known and clinically highly relevant confounders, such as sex, hypertension, or dyslipidemia; also the adjustments should use the same approach across all three cohorts to improve interpretability and readability.

Thank you for helping us clarify this aspect of the manuscript. Since the number of events in the two external populations is relatively small (n=74 in the EISNER population, n=177 in the low-dose CT population), we used backwards stepwise elimination in order to keep an event per variable ratio >10, with some authors saying 20 events may be required to ensure stable model performance (<https://www.ncbi.nlm.nih.gov/pmc/articles/PMC5394463/>). Nevertheless, we have also performed the analysis using all variables as you've suggested and present the results here.

	EISNER		Low dose CT	
	Adjusted HR (95% CI)	p-value	Adjusted HR (95% CI)	p-value
CAC 0	Reference		Reference	
CAC 1 – 100	2.27 (1.07 - 4.84)	0.033	1.68 (1.07 - 2.64)	0.023
CAC 101 – 400	5.23 (2.37 - 11.53)	<0.001	1.47 (0.90 - 2.42)	0.125
CAC > 400	6.86 (3.00 - 15.7)	<0.001	1.57 (1.03 - 2.39)	0.037
Left ventricular volume (per 10 mL)	1.25 (0.93 - 1.67)	0.138	1.05 (0.95 - 1.15)	0.360
Right ventricular volume (per 10 mL)	0.78 (0.68 - 0.91)	0.001	0.96 (0.90 - 1.02)	0.215
Left atrial volume (per 10 mL)	0.97 (0.77 - 1.21)	0.765	1.03 (0.93 - 1.13)	0.560
Right atrial volume (per 10 mL)	1.17 (0.97 - 1.41)	0.102	0.96 (0.88 - 1.03)	0.254
LV myocardial mass (per 10 g/m ²)	1.19 (0.71 – 2.00)	0.504	1.22 (1.04 - 1.44)	0.013
Shape index (per 0.1)	0.82 (0.45 - 1.47)	0.499	0.91 (0.66 - 1.26)	0.562

Eccentricity index (per 0.1)	0.61 (0.27 - 1.40)	0.245	0.92 (0.59 - 1.44)	0.707
Age (per 10 years)	1.39 (1.01 - 1.93)	0.045	1.22 (1.04 - 1.43)	0.017
Male	1.49 (0.85 - 2.59)	0.163	0.88 (0.61 - 1.28)	0.506
Hypertension	1.49 (0.91 - 2.45)	0.115	0.86 (0.63 - 1.18)	0.358
Diabetes	0.74 (0.28 - 1.91)	0.53	1.53 (1.13 - 2.07)	0.006
Dyslipidemia	0.83 (0.50 - 1.39)	0.483	0.94 (0.69 - 1.29)	0.719
Smoking	1.28 (0.54 - 3.01)	0.572	1.61 (1.12 - 2.32)	0.010
Family history	1.25 (0.93 - 1.67)	0.138	0.62 (0.34 - 1.12)	0.111

6. Also, it needs to be explicitly shown that the model works in stratified analyses for young participants and non-smokers.

Thank you for this suggestion. We have now evaluated the potential for the model to improve risk stratification in a younger population (age < 60) of non-smokers in both the EISNER and low-dose CT populations. We have included the results in the manuscript as follows:

In Results:

Risk stratification for cardiac death or MI in young (age < 60 years) non-smokers is shown in Supplemental Table 12

In Supplement:

Supplemental Table 12

	EISNER N=1240 30 (2.4%) CV death or MI	Low dose CT N=913 32 (3.5%) death or MI
AUC		
Clinical	0.731 (0.658 – 0.802)	0.707 (0.613 – 0.801)
DL - Imaging	0.793 (0.727 – 0.858)	0.739 (0.642 – 0.835)
Combined model	0.838 (0.777 – 0.898)*	0.791 (0.709 – 0.872)*
Net Reclassification Index		
Categorical	0.148 (-0.082 to 0.378)	0.263 (0.067 – 0.460)
p-value	0.208	0.009
Continuous	0.233 (-0.127 to 0.592)	0.551 (0.210 – 0.892)
p-value	0.204	0.002

Supplemental Table 12: Risk stratification for cardiac events in young (age <60) non-smokers. In the Early Identification of Subclinical Atherosclerosis by Noninvasive Imaging Research (EISNER) we evaluated cardiac (CV) death or myocardial infarction (MI). In the low dose computed tomography (CT) population we evaluated death or MI. * - $p < 0.01$ compared to clinical model, AUC – area under the receiver operating characteristic curve, DL – deep learning.

7. The Bland Altman Plots of Figure 2 are a great way to show the analyses of baseline versus one-year follow-up. Yet, the results seem to need more detailed further analyses. There is a clear non-random distribution, especially showing way greater variation in smaller mean volumes < 100 mL. In this subgroup, it seems that there are variations of more than +/- 100% between T0 and T1. As discussed earlier, the actual cut-off values of the quantiles need to be described and applied across all cohorts, this is especially also needed to help interpret the Bland Altman Results, such as it needs to be shown how many patients change risk groups between T0 and T1 and to what extent. It needs to be investigated how many patients with low scores may show a more than doubled volume in T1 and vice versa. This has a huge clinical impact and could lead to major misclassifications and potentially harm rather than benefit a patient in clinical practice.

Thank you for this suggestion. We have utilized thresholds for abnormal cardiac chamber volumes based on studies of normal subjects as outlined in response to point #3 above. We have applied those thresholds to the baseline (T0) and follow-up scans (T1) in the NLST to look at categorization of patients over time. In this analysis, we see that some patients are classified differently between time points, but the majority of patients have the same normal/abnormal classification at both points. In those patients with different classifications, the actual differences in values are still small. For example, for left atrial volume the mean difference between patients classified as abnormal at baseline but normal at follow-up was 27mL and in patients classified the opposite way 26mL. Only 5% of patients had an absolute difference > 16.9mL and 1% of patients had a difference > 43.9mL. The relevant additions to the manuscript include:

In Methods:

A comparison of patient classifications at baseline compared to follow-up imaging at one-year is shown in Supplemental Table 16.

In Supplement:

LV Volume	Normal baseline	Abnormal baseline
Normal follow-up	21997	59
Abnormal follow-up	52	78
RV Volume	Normal baseline	Abnormal baseline
Normal follow-up	21947	64
Abnormal follow-up	79	96
LA Volume	Normal baseline	Abnormal baseline
Normal follow-up	21845	66
Abnormal follow-up	113	162
RA Volume	Normal baseline	Abnormal baseline
Normal follow-up	22040	28
Abnormal follow-up	58	60
LV mass index	Normal baseline	Abnormal baseline
Normal follow-up	17792	599
Abnormal follow-up	1616	2285

Supplemental Table 16: Comparison of patient classification at baseline and follow-up imaging. LA – left atrium, LV – left ventricle, RA – right atrium, RV – right ventricle.

We also agree with the Reviewer that the Bland Altman plots are helpful in identifying patterns in the distribution of values at baseline (T0) and follow-up (T1) values. However, the previous plots may be misleading because of the size of the points, with the majority of patients represented by overlapping points. We have modified the figures to give a better sense of the actual number of outliers and the agreement in estimates between time points. The revised figure is copied here for your convenience:

Figure 2: Correlation between baseline and follow-up values.

Correlation between estimates of cardiac volumes and left ventricular mass estimated from computed tomography scans performed at baseline compared to estimated values from scans performed at 1 year in 22292 patients.

8. The authors did change the publishing license. This is good. Also, the model should be uploaded to an open-access repository and not only be available on reasonable request as this limits reproducibility.

Thank you for this suggestion. The TotalSegmentator model is available at <https://github.com/wasserth/TotalSegmentator>. We have also made the convolutional long short-term memory code available at: https://github.com/qimagingAI/CLSTM_CAC. We have now included this information in the manuscript as follows:

Data from the NLST is available through the National Cancer Institute. The TotalSegmentator code is publicly available²¹ at <https://github.com/wasserth/TotalSegmentator>, and the cLSTM code is available at https://github.com/qimagingAI/CLSTM_CAC.

9. The granularity of radiological heart assessments of NLST should be shown.

Thank you for this suggestion. Unfortunately, we do not have information regarding the specific cardiovascular abnormalities identified by the radiologist. In our analysis, we relied on the standardized codes for incidental findings, which includes only a single code for cardiovascular abnormalities. We have included this as a limitation of our analysis as follows:

In Limitations:

We have limited information regarding the exact nature of cardiovascular abnormalities which were identified. Therefore, we are not able to determine how frequently the identified abnormality was significant coronary calcification compared to other identifiable abnormalities, such as chamber enlargement or valve calcification.

Reviewer #2 (Remarks to the Author):

I commend the authors for doing an excellent and thorough job in responding to the input of both Reviewers. I appreciate that the authors have adequately addressed the concerns.

Thank you for the time and effort spent reviewing our manuscript.

1. Regarding clarification of the results relative to already available clinical variables, because CAC and structural results are so confounded by other strong predictors of mortality and CV outcomes, especially age, sex, smoking, diabetes,... that there seems to be little value in presenting/emphasizing non-risk-adjusted results like in Figures 4 and 5 (and Supplemental Figures 4 & 5).

Thank you for this suggestion. We have now moved Figure 5 to the supplemental material and decreased the discussion of these results in our manuscript. More specifically, we have deleted the following sections of text:

Compared to patients with CAC 0, CAC 1-100 (unadjusted hazard ratio [HR] 1.38, 95% CI 1.17 – 1.64), CAC 101-400 (unadjusted HR 2.16, 95% CI 1.81 – 2.58), and CAC >400 (unadjusted HR 3.33, 95% CI 2.83 – 3.92) were all at significantly increased risk.

And

Increasing CAC category was associated with an increasing risk of cardiovascular mortality. Compared to patients with CAC 0, patients with CAC 1-100 (unadjusted subdistribution hazard ratio [subHR] 1.74, 95% CI 1.13 – 2.67, $p=0.012$), CAC 101-400 (unadjusted subHR 3.62, 95% CI 2.37– 5.54, $p<0.001$), and CAC > 400 (unadjusted subHR 7.07, 95% CI 4.75 – 10.5, $p<0.001$) were all at significantly increased risk

And

Identification of cardiovascular abnormality by the radiologist was associated with less of an increase in risk (unadjusted subHR 2.07, 95% CI 1.56 – 2.73, $p<0.001$).

2. Similarly, I think the predictive value of the models evaluated here could be put in better clinical perspective by including in manuscript Figure 6 (rather than Supplemental Figure 6) the ROC for clinical models.

Thank you for this suggestion. We have now moved Supplemental Figure 6 from the supplement to the main manuscript. Additionally, we have now added similar analyses for the EISNER and low dose CT populations as outlined in response to Reviewer 1, point 2 above.

REVIEWERS' COMMENTS

Reviewer #1 (Remarks to the Author):

The authors addressed all remaining concerns in sufficient detail.